

# Dynamical changes of seismic properties prior to, during, and after 2014-2015 Holuhraun Eruption, Iceland

Maria R.P. Sudibyo[1], Eva P.S. Eibl[1], Sebastian Hainzl[1,2], and Matthias Ohrnberger[1]

[1]University of Potsdam, Institute for Geosciences, Karl-Liebknecht-Str.24/25, Potsdam-Golm, Germany
[2]GFZ German Research Centre for Geosciences, Telegrafenberg, 14473 Potsdam, Germany

**Correspondence:** Maria R.P. Sudibyo (pujiastutisudibyo@uni-potsdam.de)

**Abstract.** In volcanic eruption monitoring, it is urgent to promptly detect changes in the volcanic system during the crisis period. Ideally continuous, temporally high-resolution, multidisciplinary data is available for this. However, some volcanoes are only being monitored using a single discipline or a single seismic station. In this case, it makes sense to harvest information from the available limited data set with several different techniques. Changes in the seismic complexity could reveal the

dynamic changes due to magma propagation. We tested the performance of Permutation Entropy (PE) and Phase Permutation Entropy (PPE), which are fast and robust quantification of time series complexity, to monitor the change in the eruption process of 2014-2015 Holuhraun in Iceland. We additionally calculated the instantaneous frequency (IF), which is commonly used to monitor the frequency changes in a non-stationary signal. We observed distinct changes in the temporal variation of PE, PPE, and IF, which are consistent with the changing state from quiescence to magma propagation and then to eruption. During the

eruption, PE and PPE fit the lava discharge rate, showing their potential to forecast the duration of the eruption. Finally, we also assessed the influence of the atmospheric noise to be considered in eruption monitoring.

## 1   Introduction

A volcano should be monitored using at least four to six seismic stations (Wassermann, 2012; Saccorotti and Lokmer, 2021;

Moran et al., 2008). Yet, many volcanoes are only being monitored by fewer seismic stations (Thompson et al., 2015). Monitoring a volcano using one seismic station hinders the classification of the recorded volcano-seismic signals and their location. However, it provides the opportunity to monitor the temporal evolution of both transient and continuous seismic features, which can give an overview of the state of a volcano. Furthermore, when a volcano observatory starts operating with a single seismic station and only more seismometers are added later, seismic analysis on a single station is important to establish a continuous

baseline of monitoring.

An example of single-station monitoring is by estimating the temporal change of seismic velocity ($dv/v$), which can be influenced by magma intrusion, using ambient noise interferometry (Brenguier et al., 2008). Ambient noise interferometry can



be applied to data from a single station either by using cross-component correlation (De Plaen et al., 2016) or by autocorrelation (De Plaen et al., 2019). As a volcanic state can change quickly in a crisis, a high temporal resolution of monitoring is crucial

to do a short-term prediction. The estimation of $dv/v$ can be done in a short-time window, such as an hourly window or even minutes, however a dense network is required to achieve an accurate high temporal resolution (Illien et al., 2023). It should also be considered also that $dv/v$ changes do not necessarily relate to magma intrusions, as the physical properties of the crustal rocks can also be influenced by atmospheric pressure and temperature (Hillers et al., 2015), changes in ground water level (Sens-Schönfelder and Wegler, 2006), and ground freezing (Steinmann et al., 2020).

Entropy is a term that is broadly used to measure a level of disorder of a system, e.g. in thermodynamics (Cropper, 1986), statistical mechanics (Wehrl, 1978), and information theory (Shannon, 1948). In nature, a state that is not balanced will always shift to reach equilibrium, and this process can be associated with increasing entropy (Posadas et al., 2023). In seismology, increasing entropy can be related to the irreversible transition of unbalanced stress and strain in the crust, culminating in earthquakes (De Santis et al., 2011; Posadas et al., 2021, 2023). The changes in entropy can be linked to earthquakes' seismic

cycle (Posadas et al., 2023), where the entropy increases and reaches its maximum value at or shortly after the main shock, followed by a drop and then stabilizes during the relaxation period (De Santis et al., 2011; Posadas et al., 2021, 2023).

One of the methods to estimate entropy using the amplitude of a time series is Permutation Entropy (Bandt and Pompe, 2002). Glynn and Konstantinou (2016) calculated Permutation Entropy (PE) based on seismic time series to find a precursor prior to 1996 Gjálp eruption in Iceland. In principle, PE is estimated using one station. When several stations are available, PE can be

estimated from them to verify whether its temporal evolution is consistent at all stations. During the propagation of magma to the surface, magma will interact with the varying surrounding rocks at depth and create different seismic signals with different complexities. As magma reaches the surface, magma interaction with the shallow subsurface can generate tremor and/or long period signal. When a pre-eruptive tremor and/or long period signal is present, they exhibit a more regular oscillation and are narrow banded in frequency, and hence PE is reported to drop (Konstantinou et al., 2022).

While PE can detect a seismic precursor occurring days before an eruption (Glynn and Konstantinou, 2016), PE is also reported to be sensitive to detect fast changes. Sudibyo et al. (2022) calculated PE for 63 eruptions of Strokkur geyser in Iceland where the duration of a typical eruptive cycle is $3.7 \pm 0.9$ min (Eibl et al., 2020). The PE can resolve the typically observed four phases of Strokkur's eruptive cycle, that last several seconds to several minutes (Eibl et al., 2021), and 1 s long processes therein at high resolution (Sudibyo et al., 2022).

Apart from the amplitude, another useful property of the continuous seismic recording is the phase information. Instantaneous phase, along with the other seismic attributes, has been utilized in seismic reflection to map geological discontinuities in shallow subsurface since the 1970s (Taner et al., 1979). In seismology, the instantaneous phase has been used e.g. in seismic tomography (Bozdağ et al., 2011) and noise suppression in ambient noise cross-correlation (Schimmel et al., 2011; De Plaen et al., 2019). Kang et al. (2021) introduced the use of the instantaneous phase to calculate Phase Permutation Entropy (PPE),

which is also shown to be sensitive towards dynamic changes of a signal. Here we use the application of PE and PPE in detecting the dynamic changes before, during, and after the Holuhraun eruption 2014-2015, Iceland.





Different observable parameters from various methods are required to form a robust forecasting framework. Understanding which parameters to use and how they can represent the process in the system can help improve the framework's accuracy and reduce the computational cost. Here we also assess the derivative of the instantaneous phase, which is known as instantaneous frequency (IF)(Boashash, 1992) which is commonly used to do time-frequency analysis for a non-stationary signal.

From 16 August 2014 seismicity migrated for two weeks from subglacial Bárðarbunga volcano in North-East Iceland, first to the southeast, then to the northeast for about 48 km, at a depth of 5 to 9 km (Ágústsdóttir et al., 2016, 2019). This seismicity is interpreted as induced by a dike intrusion which is divided into 4 segments S1 to S4 (Woods et al., 2019), and culminated in an eruption that formed the Holuhraun lava flow field (see Figure 1a and b). Along the dike path, three cauldrons formed on the ice surface, possibly indicating small subglacial eruptions (Eibl et al., 2017b; Reynolds et al., 2017). After the dike reached Holuhraun, a short-lived eruption occurred during the night on 29 August 2014 and was followed by a 6-month-long eruption from 31 August 2014 at the same site. Another subglacial eruption possibly occurred on 3 September 2014 (Eibl et al., 2017b) and another subaerial eruption took place from 5 to 7 September 2014 between the ice cap and the main lava flow field (Pedersen et al., 2017; Eibl et al., 2017a). The spatial chronology is shown in Figure 1a, while the chronology timeline is shown in Figure 1b.

This 2014/2015 Holuhraun eruption is exceptionally well monitored by combining a variety of disciplines. In terms of eruption forecasting this eruption is interesting due to several subglacial eruptions, three subaerial eruptions and the extensive dike formation. In addition, a dense network of 72 seismometers (Woods et al., 2018) was distributed around the growing lava flow field providing a wealth of data. The lack of recorded shallow seismicity prior to the eruptions (Sigmundsson et al., 2015) raises a question whether the final magma movement is aseismic or generates pre-eruptive tremor as suggested by Eibl et al. (2017b).

In this paper, we first introduce the PE (section 2.2) and the PPE method (section 2.3), along with other quantifications Instantaneous Frequency (IF), RMS and RMes, and TADR. Synthetic tests to evaluate the performance of PE and PPE are provided in section 3.1. Additionally, we provide an explanation of the parameters utilized to calculate PE and PPE (section 3.2) as well as IF, RMS and RMeS (section 3.3). We then compare the temporal variation of PE, PPE, IF, with the RMS of seismic amplitude and the hypocentral distances to the station (section 4). Furthermore, we discuss the temporal variation of PE, PPE, and IF during the repose time (section 5.2), during dike propagation (section 5.3) and eruptive time (section 5.4). During the eruptive time, we compare them with the calculated lava effusive rate Coppola et al. (2017) (also section 5.4). We show the clustering of PE, PPE, and IF in section 5.5, before we provide our conclusions.

## 2 Method

### 2.1 Seismic network

A dense seismic network (network code Z7) was maintained in the area of the Holuhraun 2014-2015 eruption (White, 2010). 50 stations were running long enough to record the seismicity from the first half of 2014 to the end of 2015. Based on its proximity to the dike and eruption vents and data availability, we chose station FLUR for further analysis. This station is located about



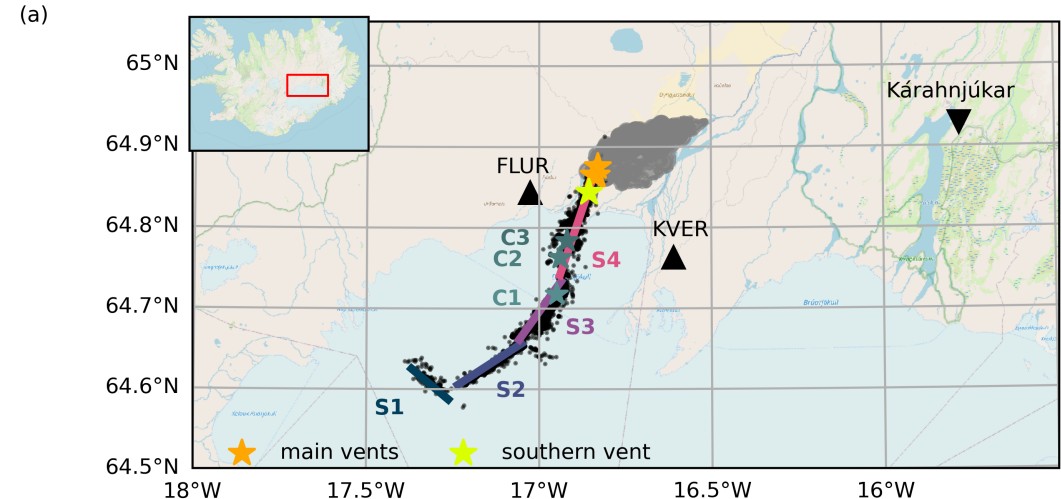

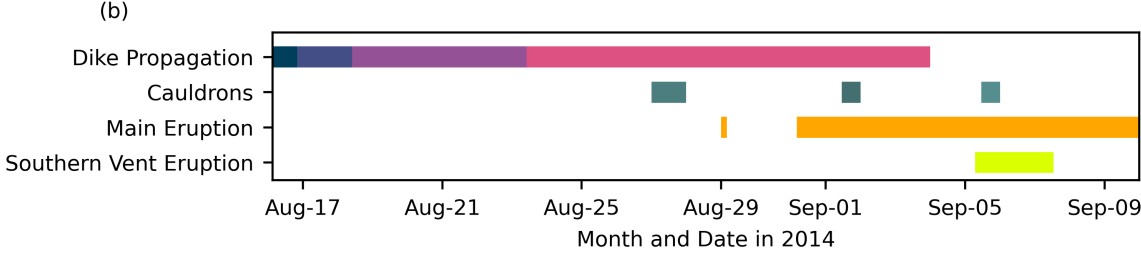

**Figure 1.** Overview of the unrest followed by the 2014/2015 Holuhraun eruption, Iceland. (a) The recorded earthquakes from the catalog of Ágústsdóttir et al. (2019) are represented by the black dots, the activated dike segments (lines; S1 to S4) as reported by Woods et al. (2019), the erupted lava flow field (gray shaded area) from Voigt et al. (2021), the location of seismic station FLUR and KVER (black triangles) and the weather station at Kárahnjúkar (black inverted triangles). (b) The temporal information of the dike propagation (Woods et al., 2019), observation of the cauldrons (Eibl et al., 2017b; Reynolds et al., 2017), the main eruption and the southern vent eruption (Eibl et al., 2017a).



32.7 km north of Bárðarbunga and about 9.1 km southwest of the Holuhraun lava flow field. At FLUR, a Guralp CMG-3ESP

broadband sensor recorded at a frequency sampling of 100 Hz. For comparison, we used station KVER which is located about

14 km southeast of the lava flow field. At KVER, a Guralp CMG-6T broadband seismometer recorded at 100 Hz sampling rate.

The location of these two stations is shown in Figure 1a.

## 2.2   Permutation Entropy (PE)

PE quantifies the probability distribution of ordinal patterns in a time window. An ordinal pattern is a vector representing the

relative order of the amplitudes. For example, a sequence of {0.32, 1.0, 2.7, 3.5, 5.0} is represented by the ordinal pattern of {0,

1, 2, 3, 4}, while {3.1, 2.2, 1.1, 3.8, 5.0} is represented by {2, 1, 0, 4, 5}. To reconstruct an ordinal pattern, we downsample the

time series using an embedding dimension $m$ and a delay time $\tau$. The embedding dimension $m$ is the total number of samples

in the sequence, and the delay time $\tau$ is the time gap between samples. The vector of the ordinal pattern is first constructed

by $[x_s, (x_s + \tau), ..., (x_s + (m-1)\tau)]$. The next ordinal patterns then are reconstructed by shifting $x_s$ one sample forward,

continuously until the last ordinal pattern reaches the end of the window.

Then, we calculate PE as:

$$PE = \frac{-1}{\log m!} \sum_{k=1}^{m!} p_k \log p_k \tag{1}$$

where $p_k$ is the probability of the ordinal pattern $k$. $p_k$ is determined by the relative frequency $N_k/N$, where $N_k$ is the number

of patterns $k$ observed in the window, and $N$ total number of ordinal patterns in the window. Equation 1 is then normalized by

the maximum number of different ordinal patterns $\log m!$, leading to a PE between 0 and 1.

An example of the PE calculation is illustrated in Figure 2. Subfigure 2a shows a 5-hour-long seismic time series recorded

by the vertical component at station FLUR, Iceland. This seismic waveform has been bandpass-filtered between 0.5-10 Hz. We

first divided the time series into 1 hour long windows without overlap. We then used $m = 7$ and $\tau = 0.04$ seconds to reconstruct

the ordinal patterns (Figure 2b). Note that the $\tau$ used in Figures 2 is for the illustration purpose while the real analysis in this

paper uses parameters mentioned in Section 3.2. Finally, in each 1 h window, we calculated its respective PE (Figure 2e).

## 2.3   Phase Permutation Entropy (PPE)

A seismic time series, denoted as $x(t)$, can be regarded as the real component of the seismic analytic trace $X(t) = x(t + iy(t)$,

where $y(t)$ is the imaginary component obtained by applying the Hilbert transform to $x(t)$ (Gabor, 1946; Taner et al., 1979;

Barnes, 1992). The Hilbert transform is an equivalent of a linear filter, where the amplitudes of a signal are unchanged but

their phases are shifted by $-(\pi/2)$ (Feldman, 2011). Scipy is a free and open source Python library (Virtanen et al., 2020)

that provides a tool to compute the analytic signal from a real signal. The function of scipy.signal.hilbert implements the step

defined by Gabor (1946) in computing an analytic signal $X(t)$, as followed:

1. Fourier transforming the real component,

2. zeroing the amplitude for negative frequencies and doubling the amplitude for positive frequencies,



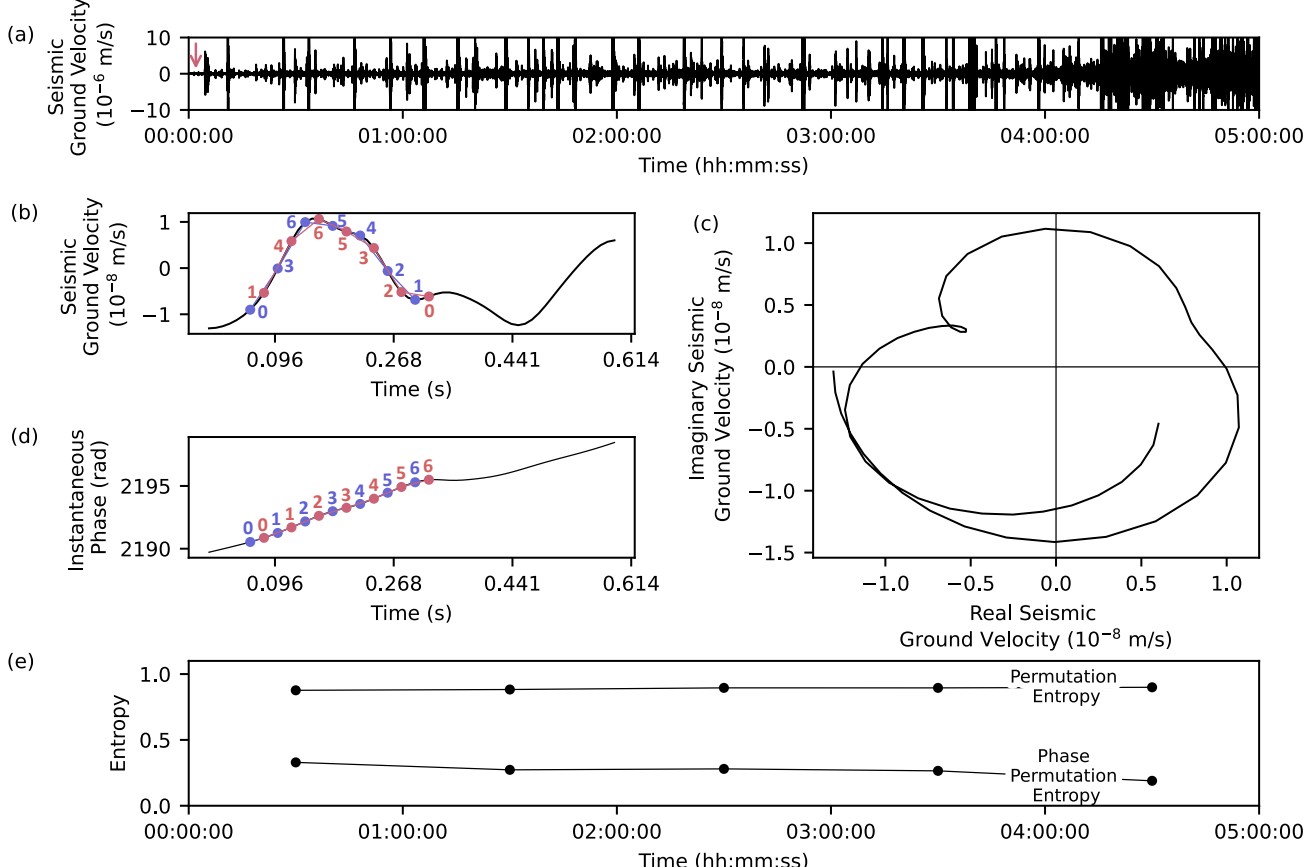

**Figure 2.** The overview of PE and PPE calculation. (a) Five hours of seismic time series from the vertical component of FLUR station, filtered between 0.5-10 Hz. (b) The amplitude of the waveform starting at the red arrow in subfigure (a) with two consecutive ordinal patterns {0, 3, 6, 5, 4, 2, 1} and {1, 4, 6, 5, 3, 2, 0}. (c) The analytic signal reconstructed from the same part of the signal in the complex plane. (d) The unwrapped instantaneous phase of the analytical signal in subfigure (c) with two examples of consecutive ordinal patterns {0, 1, 2, 3, 4, 5, 6}. Note that the linear trend of the instantaneous phase leads to fewer different ordinal patterns than the seismic amplitude data. (e) The estimated PE and PPE for the waveform in subfigure (a).





3. inverting Fourier transform.

The instantaneous phase can be defined as the degree of the $X(t)$ rotation. The instantaneous phase $\theta(t)$ is calculated using

$$\theta(t) = \tan^{-1}\left(\frac{y(t)}{x(t)}\right). \tag{2}$$

PPE is introduced by Kang et al. (2021) to calculate a wave's complexity using the instantaneous phase of an analytic signal.
In calculating PPE, we reconstruct the ordinal pattern using the instantaneous phase which is obtained from equation 2. Unlike Kang et al. (2021) who used the wrapped instantaneous phase from $-\pi$ to $\pi$ to construct the ordinal pattern, here we use the unwrapped instantaneous phase. For a reference, a sine wave will have a linear increase of unwrapped instantaneous phase, producing only one ordinal pattern. Its respective PPE value hence is 0.

An example of PPE calculation for a seismic time series (Figure 2a) is shown in Figures 2c, d, and e. We first obtain the
analytic signal (Figure 2c). We then estimate the instantaneous phase and reconstruct the ordinal patterns (Figure 2d). Finally, we estimate the PPE from the probability of the reconstructed ordinal patterns (Figure 2e).

### 2.4 Instantaneous Frequency (IF)

The instantaneous frequency (IF(t)) is defined as the derivative of the instantaneous phase $\theta(t)$,

$$IF(t) = \frac{1}{2\pi}\frac{\mathrm{d}}{\mathrm{d}t}\theta(t) \tag{3}$$

### 2.5 Root-Mean-Square (RMS) and Root-Median-Square (RMeS) of the Seismic Amplitude

The Root-Mean-Square (RMS), the root of mean squared seismic amplitude, is commonly used in volcano monitoring to continuously estimate the average seismic energy.

$$RMS = \sqrt{\frac{1}{n}\sum_i x_i^2} \tag{4}$$

In some cases where both volcano-tectonic (VT) events and tremors are present, VT events will dominate the RMS and
conceal the tremor in the RMS time series. Calculating the root of median squared seismic amplitude (RMeS) will emphasize the tremor energy more (Eibl et al., 2017a).

In this paper, RMS is calculated during the dike propagation, and RMes is calculated during the eruptive period. Both are calculated using 1h of window length without overlapping. For $x^2$ values sorted by size, RMes is calculated as follows.

$$RMeS = \begin{cases} \sqrt{x_{(n+1)/2}^2} & \text{if } n \text{ is odd} \\ \sqrt{0.5(x_{n/2}^2 + x_{n/2+1}^2)} & \text{if } n \text{ is even} \end{cases} \tag{5}$$

### 2.6 Time Average Discharge Rate (TADR)

Coppola et al. (2016) exploited the middle-infrared data acquired by multiple sensors from multiple satellites. These infrared data can be used to identify the thermal radiation emitted by volcanic eruption processes. The intensity of the thermal anomaly





is known as volcanic radiative power (VRP). During an effusive eruption, VRP can be used to calculate the time average lava discharge rate (TADR) in $m^3 s^1$.

$$TADR = \frac{VRP}{c_{rad}}, \tag{6}$$

where $c_{rad}$ ($Jm^{-3}$) is the radiant density representing the efficiency of the lava body to modulate the heat.

## 2.7 Weather data

A weather station located in Kárahnjúkar, about 59.7 km from the FLUR station (Figure 1a), provides weather data as a reference for the analysis during the repose time. The station was installed at 639 m above sea level and measures 10 m above the Earths surface. For every hour, it samples the last 10 minutes of wind speed and the last 1 minute of temperature, then calculates their average value. The data is provided by the Icelandic Meteorological Office and is available by email request.

## 3 Synthetic test and the application to seismic data

### 3.1 Synthetic tests

Sudibyo et al. (2022) tested different delay times $\tau$ to obtain optimum PE for stationary signals with different bandwidths. For a noisy stationary signal, PE is less affected by noise when $\tau$ is not too small nor equal to the fundamental period. $\tau$ is related to the frequency of the signal. When the frequency of a non-stationary signal changes, it will influence the calculated PE as the calculation uses a fix $\tau$. This behavior was not investigated in detail in the previous study.

Here we investigated the differences in PE and PPE calculation on three different synthetic signals. First, we generated a chirp signal with increasing frequency from 1 to 10 Hz and a sampling rate of 400 Hz (Figure 3a) and then added a demeaned uniform noise (Figure 3d). Furthermore, we also tested the noise itself (Figure 3g). We used the Numpy library (Harris et al., 2020) to create the chirp signal and the noise, and then calculated their PE and PPE for every 1s-window length. We use $m = 5$, which is the highest $m$ possible to calculate PE and PPE based on the number of samples for 1s window length.

In this test, we focus on how a changing frequency could affect the PE and PPE calculation. We tested different $\tau$ from the shortest possible to 0.5 s, which is the Nyquist period of the high-frequency corner of the signal. Figures 3b and e show that using $\tau > 0.25$ s, which is half of the Nyquist period, generates artifacts in PE of the higher frequencies. Figure 3b shows that the PE of the chirp signal increases when the frequency increases. This is due to the less repeating ordinal patterns reconstructed when the frequency increases. An opposite behavior is shown in Figure 3e for the noisy chirp. Noise increases the PE (Figure 3h). Due to the fix $\tau$, the ordinal pattern samples relatively more noise at lower frequencies. When the frequency increases, the length of the ordinal patterns get closer to the signal's wavelength and the signal-to-noise ratio increases, resulting in a lower PE at higher frequency.

PPE utilizes the instantaneous phase. The unwrap instantaneous phase of a chirp signal is always increasing (Figure S1a), producing only one ordinal pattern, resulting in PPE=0. In the case of the noisy chirp, the noise contains non zero-crossing





**Figure 3.** Synthetic test (a) time series of a chirp signal, (b) PE and (c) PPE. (d-f) Same as subfigures (a-c) for a noisy chirp signal. (g-i) Same as subfigures (a-c) for noise.

amplitude and generates tangled rotation in its analytical signal as shown in Figures 2c, and S1f and j. These tangled rotations have decreasing phase angles which generate different ordinal patterns (Figures S1h and i), hence increasing PPE.

A common practice to reduce noise and isolate the signal of interest is by filtering. We tested how PE and PPE change if a noise gets filtered. Figure S2 shows a decrease in PE and PPE in the filtered noise, with wider bandwidth possessing higher PE and PPE than a narrower bandwidth.

### 3.2   Parameter selection in calculating PE and PPE of seismic time series

The parameters required to calculate PE and PPE are window length, embedding dimension $m$, and delay time $\tau$. To determine
the window length, we need to fulfill two requirements, the minimum number of samples required by the embedding dimension





$m$ and the targeted resolution in the temporal variation of PE and PPE. To calculate PE and PPE, the number of samples in a window has to be more than $m!$, where $m!$ is the maximum number of possible ordinal patterns reconstructed from the embedding dimension $m$. In the case of a noisy signal, it is advised to use window length longer than 5! (Amigó et al., 2008) to cover all possible patterns generated by the noise. Following the suggestion, for $m = 7$ we need a minimum sample of 25200 points or equal to 252 seconds. As seismic precursors can range from hours to days prior to an eruption, we chose 1 hour of window length which also fulfills the requirements of the number of samples needed by the mentioned embedding dimension. The delay time $\tau$ needs to be smaller than the Nyquist period of the targeted signal (Berger et al., 2017), and our synthetic test recommends $\tau < 0.5$ Nyquist period. Our interest lies in frequency between 0.5 to 10 Hz. Tremor is found to be between 0.8 to 2.5 Hz (Eibl et al., 2017b) and we found regular repeating noise at frequencies above 10 Hz. The Nyquist period for a 10 Hz signal is 0.05 s, and we chose $\tau = 0.02$ s.

We processed 2 years of seismic data recorded at the vertical component of station FLUR from January 2014 to December 2015, covering the repose period, the unrest and the eruption. We used Obspy to read the seismic data and to apply a Butterworth bandpass filter of order 4. A Butterworth filter does not create ripples in the pass-band, which is important to avoid artificial ordinal patterns. We activated the option zero-phase in the Butterworth filtering, to obtain no phase shift in the filtered seismogram. All plotting is done using Matplotlib. A comparison between the two stations, FLUR and KVER, shows similar temporal variations of PE and PPE in both stations (Figure S3).

### 3.3 Calculation of IF, RMS, and RMeS

We also used the vertical component of FLUR for the IF, RMS, and RMeS calculations and adapted the same 0.5-10 Hz frequency band used for the IF calculation. First, we estimated the Instantaneous Phase for every seismic data sample and then we calculated IF from every two consecutive Instantaneous Phases. We then calculate the mean IF for every 1h window, to obtain the same resolution as PE and PPE. The calculation of RMS and RMeS also uses the same 1h window length, and within the frequency from 0.5 to 10 Hz.

## 4 Results

In the following, we will describe the temporal variation of various features of the seismic waveform in the last 5 days of the repose time, the 14 days of earthquake migration, and the first 15 days of the main eruption (Figure 4). We will also describe the further evolution until March 2015 when the eruption ended (Figure 5).

### 4.1 The earthquakes hypocentral distances and RMS of seismic amplitude

More than 30,000 earthquakes are listed in the earthquake catalog by Ágústsdóttir et al. (2019). They migrate from the southeastern rim of subglacial Bárðarbunga volcano at 29-32 km distance and then progress to the Holuhraun lava field at about 8-12 km distance from station FLUR (Figures 1a and 4a). The dike segments S1 to S4 (Woods et al., 2019) feature distinctly different distance ranges. The hypocentral distance to our station decreased quickest at the beginning of each segment when it





**Figure 4.** Overview of the seismic characteristics during the quiescence, eruption preceding seismicity, and the first 15 days of the eruption: (a) hypocentral distances of the earthquakes to the FLUR station, (b) RMS, (c) PE, (d) PPE, and (e) mean IF. RMS, PE, PPE, and IF are calculated from seismic time series recorded at the vertical component of station FLUR, filtered between 0.5-10 Hz. The segmentation of the dike (S1-S4, dark blue to pink horizontal lines), cauldrons observed on glacier surface (C1-C3, dark green horizontal lines), tremor on 03.09.2014 interpreted as presumed subglacial eruption (Eibl et al., 2017b) (light blue), eruption from the southern vent (light green horizontal line and light green star) and the main eruption (yellow horizontal and yellow stars) are marked in subfigure (a) and by dashed vertical lines in all panels.





formed and the distance stalled towards the end of a segment. In segment S3, the earthquakes moved in several episodes, on 18-20 August 2014, and later a short quiescence occurred on 22 August 2014 at 10:00. While in segment S4, the earthquakes kept moving to the northeast, their distances to FLUR remained around 10 km. After the onset of the main eruption on 31 August 2014, the number of earthquakes became fewer and the seismic time series was dominated by eruptive tremor.

Consequently, the seismic waveform is not only dominated by earthquakes but also features seismic volcanic tremor. During repose time the RMS is mostly below $5 \cdot 10^{-7}$ m/s. During dike formation, the RMS of the seismic amplitude is very spiky and affected by the earthquakes (Figure 4b). RMS increases significantly after S4 starts on 23 August 2014 and is also dominated by spikes throughout the segment and the eruptive period. During S4, the RMS exhibits few fluctuations from 23-24 August, 24-26 August, 26 August-1 September 2014. During the eruptive period, there is an increase between 2-4 September. Afterward, the RMS amplitude is mostly low and decreases with time with spikes throughout the eruptive period.

## 4.2 The temporal evolution of Permutation Entropy (PE)

During the repose time, PE displays a strong daily variation (Figure 4c). A sharp increase is observed on 16 August 2014 when the earthquakes occur and start to migrate. PE stays high, mostly above 0.6 during the two weeks of the earthquake migration. Prior to the main eruption on 31 August 2014, PE exhibits three decreasing trends: segment S1 to S2, S2 to S4, and a more gentle slope during S4 to the onset of the main eruption. After the main eruption begins, there is a notable decrease in PE, followed by a strong drop on 3 September 2014, when PE reaches a value of 0.37. The drop on 3 September represents a local minimum that persists for one day. PE then fluctuates between 4 and 7 September and subsequently gets more steady after 10 September 2014. PE generally declines toward the end of the eruption (Figure 5a). The comparison of PE with TADR from Coppola et al. (2017) reveals a similar shape of both (Figure 5a). Note that the TADR is plotted on a log scale, while PE is in a linear scale.

## 4.3 The temporal evolution of Phase Permutation Entropy (PPE)

Similar to PE, PPE also exhibits a strong daily variation during the repose time (Figure 4d). Interestingly, during the earthquakes migration, PPE follows a pattern that is anti-correlated to that of PE and IF. PPE increases from 0.27 at S1 to 0.32 at S2, but followed by a drop to 0.224 in the next two hours. PPE then drops to 0.14 on 20 August followed by a sharp increase until 23 August 2014. An abrupt drop occurs on 23 August 2014 when the time segment S4 starts, followed by two successive increasing trends which culminate in the eruptions on 29 and 31 August 2014. Before the eruption, another peak is observed on 22 August 2014 at 10:00 and 26 August at 07:00. These peaks occurred one day before the presumed subglacial eruptions on 23 and 27 August 2014. PPE increases abruptly right after the main eruption starts. Interestingly, this is then followed by a pattern that is similar to the pattern in PE. PPE also drops and reaches the minimum value on 3 September 2014 for 1 day and generally declines towards the end of the eruption. The trend observed in PPE also aligns well with the shape of the lava effusion rate (Figure 5b), with PE plotted in a linear scale and TADR in a log scale.





**Figure 5.** Overview of the 6-month long subaerial eruption. Comparison of (a) PE, (b) PPE, (c) mean IF and (d) RMeS seismic velocity (all black) on a linear y-axis, with lava effusion rate Coppola et al. (2017) plotted on a logarithmic y-axis (orange line, scale on right). Note the different y-scales for PE and PPE. All values were calculated using 1 day window-length. The dashed line marks the reference line, obtained from the value of PE, PPE, IF, and RMeS on 8 September 2014, when the eruption starts to stabilise.





## 4.4 The temporal evolution of the mean Instantaneous Frequency (IF)

Before the start of the two-week migration of earthquakes, the mean IF is mostly low between 0.5 to 0.8 Hz and exhibits a daily
variation (Figure 4e). An abrupt increase in mean IF is observed when the swarm starts on 16 August 2014. IF is generally
increasing from 16 to 20 August 2014, with values from 3 to 5 Hz during the propagation of segments S1 to S3. Even though
it generally increases, IF exhibits a quasi-periodic fluctuation with each period varying from 4 to 12 hours. There are two
decreasing trends within this fluctuation, which fit the time length of segment S1 and S2. IF was the highest at the beginning
of both segments and continuous dropping towards the end of the segments. IF is generally higher during S2 than during S1.
IF continues to increase from the beginning of S3 for two days before decreasing towards the beginning of S4. During the
propagation of S4, IF decreases fast before the onset of the eruption on 29 August and continues a more slow decrease towards
the main eruption on 31 August. During the earthquakes' migration, IF is anti-correlated to PPE.

Two sharp drops of IF are also observed at 10:00 on 22 August and at 07:00 on 26 August 2014, at the same time when the
two peaks of PPE are observed. Another drop is also observed on 28 August 2014 at 22:00 followed by a sharp increase to the
260 maximum value of 5.6 Hz, marking the onset of the short eruption on 29 August 2014. After the onset of the main eruption on
31 August 2014, IF drops from 4.2 Hz to 2 Hz. IF increases slightly from October to November 2014, then in general decreases
to the end of the eruption on 28 February 2015 (Figure 5c).

## 5 Discussion

### 5.1 Factors affecting the complexity quantification in PE and PPE

The synthetic tests in subsection 3.1 show that PE, calculated using a fix $m$ and $\tau$, is higher for a signal with higher frequency
content compared to one with lower frequencies. A signal with energy in a broad frequency bandwidth usually possesses a high
PE (Dávalos et al., 2021; Sudibyo et al., 2022). Independent of its frequency content, PPE is more affected by the presence
of non-zero-crossing oscillation in its signal. When a low-frequency signal is superposed with a weaker signal with a higher
frequency, most oscillations of the higher-frequency part do not cross zero amplitude. This oscillation type will cause more
complex rotation in its analytical signal (Figure S1f), resulting in more ordinal patterns and higher PPE.

### 5.2 The influence of atmospheric processes on PE, PPE, and mean IF

We investigated the influence of temperature and wind on the hourly PE, PPE, and mean IF during different seasons from
2014 to 2015. Seasonal variation including temperature, wind speed, and air pressure is common to be observed in seismic
ambient noise (Bormann and Wielandt, 2013). Atmospheric pressure and temperature can generate noise at frequencies below
50 mHz (Bormann and Wielandt, 2013), while wind can generate noise at higher bandwidth between 0.5 to 60 Hz (Bormann
and Wielandt, 2013; Withers et al., 1996). Here we observe strong effects of atmospheric signals on the seismic characteristics
in the repose time before the magma propagation, and after the eruption. It should be noted that the seismic station FLUR





and the weather station Kárahnjúkar are about 57.9 km apart. While we noticed a shift between the variation of wind speed in comparison with PE, PPE, and IF about 1-2 hours in the hourly window, they can be considered negligible.

Wind speed is found to be strongly correlated with the background noise at frequency higher than 1 Hz (Withers et al., 1996). We observed the strong influence of wind speed on the RMS of the seismic amplitude during the whole repose period (Figures S5h-S12h). Interestingly, we did not observe the influence of wind on PE, PPE, and mean IF in summer. The influence of wind speed on PE, PPE, and mean IF could be seen clearly in spring, especially from February to May (Figures S5 e-g, S9 e-g, S10 e-g) when the fastest wind speed reaches 40 m/s in March 2014, in autumn from September to October (Figure S11 e-g), and

in winter (Figure S12 e-g). Wind seems to already influence PE and PPE at the end of the eruption from January to February 2015 when the wind speed reaches about 25 m/s and the tremor amplitude is low, while the mean IF is less affected (Figures S8 e-g and S9 e-g). Nevertheless, the wind effects on PE, PPE, and IF are found to be negligible during the magma propagation and the main eruption phase.

During the repose time, PE, PPE, and mean IF show clear daily cycles with high correlations with the temperature changes,

as shown in Figure S6 a-c, for the temperatures recorded by the Icelandic Meteorological Office between June and August 2014. Similar results are observed for the summer months of 2015 (Figures S10 a-c and S11 a-c). This is surprising because the influence of temperature on seismic wave with frequencies higher than 50 mHz is usually very small and negligible for most seismological processing. For example, it does not seem to influence the temporal variation of the RMS of seismic amplitude for the whole observation period (Figures S5d-S12d). However, PE and PPE depend only on the order of the consecutive

values of the amplitude and instantaneous phase, but not on their magnitudes. A very small difference in the consecutive values can change the order and create a different ordinal pattern, hence increasing the calculated entropy. Temperature affects both the thermoelasticity of the seismometer, especially the analyzed vertical component (Bormann and Wielandt, 2013), and the underlying rocks (Prawirodirdjo et al., 2006). However, the temperature effect on PE, PPE, and mean IF are found to be negligible during the magma propagation and the main eruption.

Donaldson et al. (2019) estimate over 10 years variation of relative velocity changes ($dv/v$) in the crustal rock of central Iceland, including Bárðarbunga and Holuhraun. They observed the $dv/v$ to be high in the winter and spring and low in the summer and fall. This seasonal variation is associated with (i) the changes in the elastic loading on the rocks due to the seasonal changes of snow thickness and the atmospheric pressure, (ii) the annual variation of the ground water level. They did not compare $dv/v$ with the wind speed and temperature. Wind speed is usually not considered to influence $dv/v$, while

the atmospheric temperature can still affect the $dv/v$ through the thermoelasticity of the crustal rocks (Hillers et al., 2015; Prawirodirdjo et al., 2006). In our two-year observation, we did not observe a clear change due to these long-period seasonal variations. While these seasonal changes in the crustal properties might affect the variation of PE, PPE, and IF, it seems to be much weaker compared to the daily variation due to the atmospheric noise.

### 5.3   The influence of the magma propagation on PE, PPE, and IF

PE and mean IF increase sharply on 16 August 2014. Both values remain at elevated levels until the onset of the main eruption on 31 August 2014. In contrast, PPE does not increase, but its fluctuation gradually decreases until the main eruption. The high





PE and mean IF are caused by the high dominant frequencies of earthquakes and their energy distribution in a broad frequency range. We investigated the rotation of their analytic signal in a complex plane (Figure S4b). Compared to ambient noise (Figure S4a), earthquakes exhibit less complex rotation and therefore lower PPE.

PE and mean IF increase not only at the initial start of the magma propagation (segment S1), but also at the start of other segments, followed by a gentle, gradual decrease towards the end of each segment (Figure 4a,c). Only for S3 does PE not show this pattern, while mean IF shows it for all segments. This pattern may reflect that the magma propagation in the initial phase of each segment released more fracture energy to open the pathways, and less energy was needed later when the dike only continued to open until the dike extended into another segment (Sigmundsson et al., 2015; Ágústsdóttir et al., 2016, 2019). We

checked the distribution of the earthquakes' magnitudes and observed that the magnitudes are mostly smaller at the end of the segments. However, the RMS signal is too spiky to clearly show the same trend (Figure 4d). An equivalent case is observed in the seismic cycle of tectonic earthquakes, where the Shannon Entropy is found to gradually drop to its initial state after the main shock during the post-seismic states (De Santis et al., 2011; Posadas et al., 2023). Here, the main shock corresponds to the dike migration phase in a segment, and the post-seismic state corresponds to the dike thickening after its extension.

PPE of earthquakes is found to be lower than the PPE of the ambient noise. Its temporal evolution during the earthquake migration exhibits stronger changes than PE. At the beginning of segments S1 to S4, PPE is low and then increases to the end of the segment. As most of the seismic moment is released and more earthquakes are generated at the beginning of each dyke segment, PPE first drops and then increases towards the end of the segment when the number of earthquakes becomes less. During segment S3, PPE did not drop suddenly, but decreased over 2 days until 20 August 2014, before increasing to the

end of the segment. An anti-correlated trend is shown by the mean IF, while it increased from 18 to 20 August 2014 before it decreased back to the end of segment S3. From 19 to 23 August 2014, the earthquake migration was reported to stop before changing its direction from south-east to north-east (Sigmundsson et al., 2015).

A single peak of PPE and a drop of mean IF are observed on 22 August 2014 at 07:00 to 13:00. These are associated with the short period of quiescence during this time. When the S4 dike segment formed, the earthquakes reached the closest distance

from FLUR station, which is about 10 km (Figure 4e). These earthquakes dominate the time series, causing PPE to reach the lowest trend, while the RMS reaches the highest value (Figure 4d).

After stopping for 81 hours, the dike started to move again and generated segment S4 on 23 August 2014, it was accompanied by a pre-eruptive tremor (Eibl et al., 2017b). This tremor could also be associated with the formation of cauldron C1 which was visually observed on 27 August 2014 (Reynolds et al., 2017). However, this tremor was concealed by a high seismicity

rate during the lateral movement of segment S4 (Figure 1). Therefore, it is not seen in PE, PPE and mean IF.

There was a lack of shallow seismicity prior to the eruption (Ágústsdóttir et al., 2019; Eibl et al., 2017b). Following the earthquakes moving horizontally at depth 5 to 8 km (Ágústsdóttir et al., 2019; Woods et al., 2019), long period (LP) events were detected for about 10 days, starting from 25 August 2014. These LP events had a dominant frequency ∼1Hz, clear P and S onset, and occurred at ∼4 km depth (Woods et al., 2018). Eibl et al. (2017b) suggested the possibility of pre-eruptive

tremor that is formed by repetitive microearthquakes at less than 3 km depth followed by silent magma migration to the surface. However, by utilizing PE, PPE and mean IF, we could not observe any changes that could be related to pre-eruptive tremor



before the eruption onsets on 29 and 31 August 2014. The seismic wave generated by earthquakes seems to be dominating the time series, masking other processes, especially during the last dike segment S4, when the earthquakes reach the closest distance to the station, and the changes in PE, PPE, and mean IF become less significant than the earlier segments.

Eruption forecasting is easier when the pre-eruptive process generates a pattern of distinct seismic events that are chronologically changing in time and depth. In the case of Strokkur Geyser, the PE variation can characterize the four different phases in the geyser's eruptive cycle (Sudibyo et al., 2022), which are eruption, conduit refilling, gas accumulation in the bubble trap, and the collapses of the bubble gas in the shallow conduit (Eibl et al., 2021). These different processes generate distinct signals with different complexity, thus resulting in distinct values of the corresponding PE. Furthermore, each phase takes place in sep-
arated locations and depths (Eibl et al., 2021). Therefore, not only Sudibyo et al. (2022) can observe a high correlation between PE and the hypocentral distance between seismic event's source and the seismic station, they can also use PE to accurately predict the geyser's eruptions. Konstantinou et al. (2022) reported a consistently decreasing PE prior and during 3 eruptions in Shinmoedake, a stratovolcano in Japan, which is associated to the dominant occurrence of the pre- and eruptive tremor. In Shinmoedake, as magma moves to the shallower depth and the higher frequency of the seismic events get attenuated, PE drops
before the eruption starts. In contrast, the 2014/2015 Holuhraun eruption is preceded by two weeks of lateral dike propagation, dominated by high-frequency events. Changes in the types of seismic events is minor during the pre-eruptive process, and the majority of the events do not move to the shallower depth, causing less significant evolution in the seismic parameters to the eruption's onset.

## 5.4   PE, PPE, and mean IF reflecting the dynamics of eruptive tremor

In contrast to the pre-eruptive tremor, the dynamic of the eruptive tremor is well reflected by the properties of PE, PPE, and mean IF. After the main eruption starts, the eruptive tremor dominates the 6 months of eruption, and we observed decreasing values of PE and mean IF, while PPE increases. Volcanic tremors have been reported to have a low PE due to their low dominant frequency and narrow spectral distribution (Konstantinou et al., 2022). The eruptive tremor in Holuhraun has most energy in a low and narrow frequency band ranging from 0.8 to 2.5 Hz (Eibl et al., 2017b). We investigated the seismic analytic wave of
the eruptive tremor and found that its rotation is more complex compared to earthquakes (Figure S4d). While the presence of noise could increase the complexity of phase angle rotation, the trend of PPE does not align with the trend of the ambient noise (Figures S7 and S8. Therefore, it is more likely that the calculated PPE represents the characteristics of the eruptive tremor itself.

    Eibl et al. (2017a) have identified multiple sources generating the eruptive tremor accompanying the Holuhraun eruption.
These sources are associated with fissure locations and the height of the lava fountain, the growth of the lava flow field and intrusions at depth. Hibert et al. (2015) found a linear correlation between seismic energy of tremor with lava effusive rate in Piton de la Fournaise volcano in La Réunion island. Figure 5 shows the comparison of the magma effusive rate estimated by Coppola et al. (2017) with the temporal variation of PE, PPE, mean IF, and seismic root-median-square (RMeS), where we fitted a horizontal dashed line to the early values of each parameter as a reference and observed decrease of the values with
time. Both PE and PPE show an alignment with the magma effusive rate (Figure 5 b and c). Starting from January 2015, PE





and PPE start to decline faster, which indicates the eruption is ending. A similar pattern was observed by Sudibyo et al. (2022), studying the temporal variation of PE during 63 eruptions of Strokkur Geyser, Iceland. During eruptions with two to four water fountains in quick succession, PE stays high and only drops after the end of the last water fountain of one eruption.

On 3 September 2014, a strong tremor was recorded for about 21 hours (Eibl et al., 2017b; Woods et al., 2018) which is assumed to have preceded a sub-glacial eruption deepening either the cauldron C2 (Eibl et al., 2017b) or C1 (Woods et al., 2018). We noticed the drop in PE and PPE on 3 September 2014. PE and PPE reach the minimum value at 12:00 for 6 hours before they start increasing back at 18:00. The analytic signal shows fewer entangled rotations (Figure S4c) than the main eruption tremor, which is similar to earthquakes (Figure S4b) and causes low PPE. The PE and mean IF are also low, suggesting that the energy of the tremor is concentrated in a low and narrow frequency band. This result supports Eibl et al.

(2017b), who suggested that the tremor is comprised of swarms of microseismicity associated with the fracturing of the shallow crust above the dike.

Woods et al. (2018) observed that the tremor on 3 September has a similar spectral content as the LP swarms which were recorded from 25 August to 2 September 2014. They suggested that the LP swarms could represent magma moving above the dike which culminates in tremor, producing the sub-glacial eruption. As we cannot resolve the LP swarms during the mentioned

period, we cannot confirm nor reject their interpretation.

PE, PPE, and mean IF then undergo a fluctuation from 4 to 7 September 2014 (Figures 4 b and c). This fluctuation could be associated with the opening of two fissures located on the south of the main eruption, resulting in a minor eruption from 5 to 7 September 2014. Thereafter, the PE, PPE, and mean IF started to be more stable throughout the eruption (Figures 5 a, b and c).

### 5.5 Cluster analysis

Based on the chronology information, we plotted PE, PPE, and mean IF into four clusters: quiescence, dike segment formation S1 to S4, the presumed subglacial eruption, and the subaerial eruption. This aims to illuminate whether the different clusters are separated in space or overlap each other. Figure 6 shows a good separation between the clusters. While having a similar trend, the S1 and S2 segments are also separated from S3 and S4 in all plots. This result endorses the potential of PE, PPE, and IF capability to discriminate events and monitor the eruption process.

## 6 Conclusions

In this study, we assessed the capability of PE, PPE, and IF to characterize the changing state before, during, and after the 2014-2015 Holuhraun eruption in Iceland, by utilizing continuous seismic time series. We observed that temperature and wind influence strongly influence PE and PPE during the repose time, but their effect is minor during the dike migration and eruption. We found that PE, PPE, and IF can resolve the different stages from the repose time followed by the dike propagation through

different segments, the multiple eruptions during the eruptive period, and the end of the main eruption. We show that combining these parameters could be a useful tool in discriminating different seismic signals and monitoring their evolution in the time



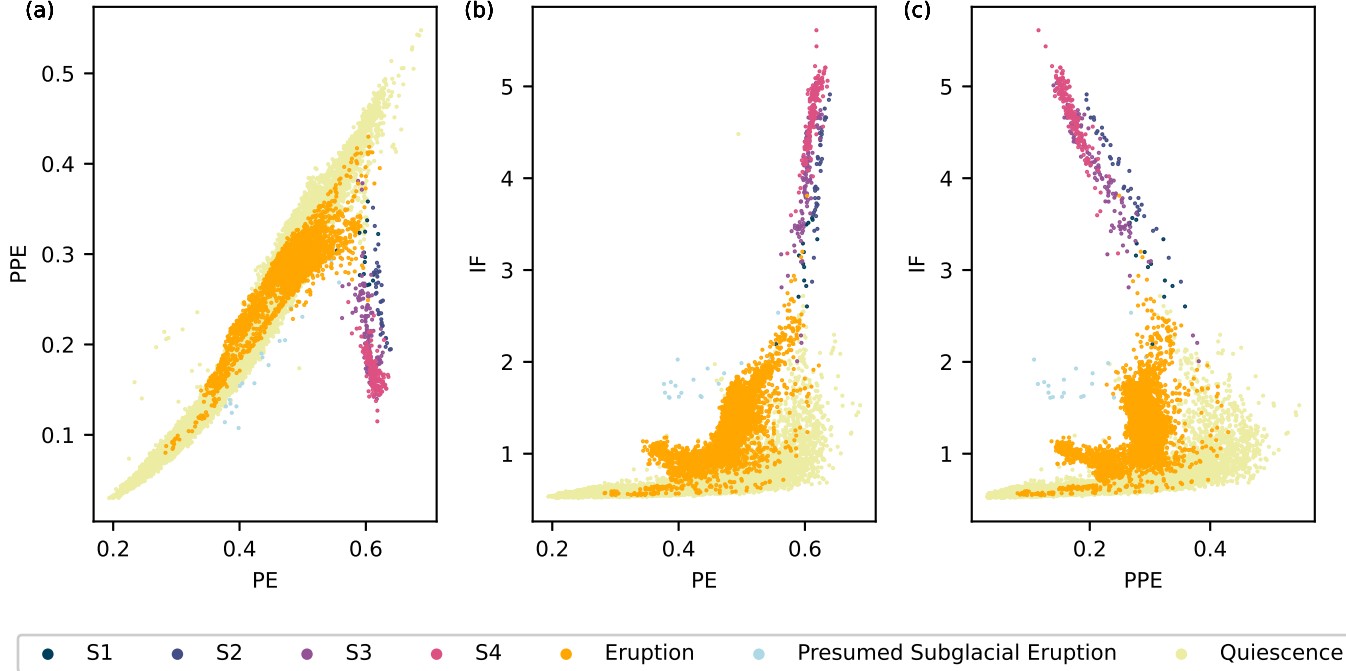

**Figure 6.** Clustering plots: (a) PPE vs. PE, (b) IF vs. PE, and (c) IF vs. PPE for the quiescence (black), dike segments S1 to S4 (blue to dark orange), and the eruption period (orange).

domain. While our study shows that it is still challenging to determine the eruption onset after the start of the pre-eruptive seismicity, PE and PPE have the potential to help predict the duration of the eruption and when the eruption is likely to end.

*Data availability.* The seismic waveform data of FLUR and KVER stations are part of Northern Volcanic Zone (NVZ) seismic network
installed by the University of Cambridge, United Kingdom. The data is publicly available via the IRIS Data Management Center (IRISDMC).

*Author contributions.* Conceptualization: MRPS, EPSE, SH; Formal Analysis: MRPS; Funding acquisition: MRPS, EPSE, SH; Investigation: MRPS; Supervision: EPSE, SH; Visualization: MRPS; Writing – original draft preparation: MRPS; Writing – review and editing: MRPS, EPSE, SH, MO

*Competing interests.* The authors declare that they have no conflict of interest



*Acknowledgements.* We thank the University of Cambridge which made the seismic data of Northern Volcanic Zone (NVZ) seismic network publicly available. We thank Guðrún Nína Petersen from the Icelandic Meteorological Office for providing the weather catalog on 19 November 2020. This research is funded by DAAD Doctoral Research Grant 57507871.



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
