# Peer review of "Dynamical changes of seismic properties prior to, during, and after 2014-2015 Holuhraun Eruption, Iceland"

_EGUsphere, 2024_

## Author Comment (AC1)

Figure 1: Clustering plots based on K-means algorithm in a 3-D space formed by PE, PPE and mean IF (a-c) and based on expert interpretation(d-f). Cluster 1 is associated with quiescence, cluster 2 is associated with dyke segments S1 to S4, and cluster 3 is associated with the eruption and presumed subglacial eruption. The border between cluster 1 and 3 by K-means are noticeably different compared to the border between quiescence and eruption according to the expert interpretation.

| K-Means
Expert Interpretation          | Cluster 1 | Cluster 2              | Cluster 3 |
|-------------------------------------------|-----------|------------------------|-----------|
| Quiescence                                | 0.950     | $8.298 \times 10^{-5}$ | 0.049     |
| Dyke Propagation (S1-S4)                  | 0         | 0.964                  | 0.036     |
| Eruption and presumed subglacial eruption | 0.232     | 0.013                  | 0.755     |

Table 1: Confusion matrix between clusters formed by K-Means and the expert interpretation in Figure 1. The first row shows that 95% of the data points during quiescence are classified into cluster 1 and 5% are classified into cluster 3 and 2. The second row shows that 96% of the data points during the dyke propagation are classified into cluster 2 and 4% are classified into cluster 3. The third row shows that 23% of the data points during the eruption are classified into cluster 1, 76% are to cluster 3 and 1% are to cluster 2. The summation of values in each row is equal to 1.

1

---

## Author Comment (AC2)

[Figure]

Figure 1: Clustering plots based on K-means algorithm in a 3-D space formed by PE, PPE and mean IF (a-c) and based on expert interpretation(d-f). Cluster 1 is associated with quiescence, cluster 2 is associated with dyke segments S1 to S4, and cluster 3 is associated with the eruption and presumed subglacial eruption. The border between cluster 1 and 3 by K-means are noticeably different compared to the border between quiescence and eruption according to the expert interpretation.

| K-Means
Expert Interpretation | Cluster 1 | Cluster 2 | Cluster 3 |
|---|---|---|---|
| Quiescence | 0.950 | $8.298 \times 10^{-5}$ | 0.049 |
| Dyke Propagation (S1-S4) | 0 | 0.964 | 0.036 |
| Eruption and presumed subglacial eruption | 0.232 | 0.013 | 0.755 |

Table 1: Confusion matrix between clusters formed by K-Means and the expert interpretation in Figure 1. The first row shows that 95% of the data points during quiescence are classified into cluster 1 and 5% are classified into cluster 3 and 2. The second row shows that 96% of the data points during the dyke propagation are classified into cluster 2 and 4% are classified into cluster 3. The third row shows that 23% of the data points during the eruption are classified into cluster 1, 76% are to cluster 3 and 1% are to cluster 2. The summation of values in each row is equal to 1.

[Figure]

Figure 2: Clustering plots based on K-mean algorithm in a 4-D space formed by PE, PPE, IF and RMS (a-f) and based on expert interpretation(g-l). Cluster 1 is associated with quiescence, cluster 2 is associated with the eruption and presumed subglacial eruption, and cluster 3 is associated with dyke segments S1 to S4.

| Expert Interpretation \ K-Means | Cluster 1 | Cluster 2 | Cluster 3 |
|---|---|---|---|
| Quiescence | 0.953 | 0.046 | $8.298 \times 10^{-5}$ |
| Dyke Propagation (S1-S4) | 0 | 0.036 | 0.964 |
| Eruption and presumed subglacial eruption | 0.234 | 0.744 | 0.022 |

Table 2: Confusion matrix between clusters formed by K-Means and the expert interpretation in Figure 2. The first row shows that 95% of the data points during quiescence are classified into cluster 1, and 5% are classified into cluster 2 and 3. The second row shows that 96% of the data points during the dyke propagation are classified into cluster 3, and the 4% are classified into cluster 2. The third row shows that 75% of the data points during the eruption are classified into cluster 2, 23% are classified to cluster 1 and 2% are classified to cluster 3. The summation of values in each row is equal to 1.

[Figure]

Figure 3: Same as Figure 2 but with RMeS instead of RMS. Cluster 1 is associated with quiescence, cluster 2 is associated with dyke segments S1 to S4, and cluster 3 is associated with the eruption and presumed subglacial eruption.

| K-Means Expert Interpretation | Cluster 1 | Cluster 2 | Cluster 3 |
|---|---|---|---|
| Quiescence | 0.950 | $8.298 \times 10^{-5}$ | 0.049 |
| Dyke Propagation (S1-S4) | 0 | 0.964 | 0.036 |
| Eruption and presumed subglacial eruption | 0.231 | 0.012 | 0.756 |

Table 3: Confusion matrix between clusters formed by K-Means and the expert interpretation in Figure 3. The first row shows that 95% of the data points during quiescence are classified into cluster 1 and 5% are classified into cluster 2 and 3. The second row shows that 96% of the data points during the dyke propagation are classified into cluster 2 and 4% are classified into cluster 3. The third row shows that 23% of the data points during the eruption are classified into cluster 1, 76% are classified to cluster 3 and 1% to cluster 2. The summation of values in each row is equal to 1.

[Figure]

Figure 4: Same as Figure 2 but with log(RMS) instead of RMS. Cluster 1 is associated with quiescence, cluster 2 is associated with eruption and presumed subglacial eruption, and cluster 3 is associated with dyke segments S1 to S4.

| K-Means
Expert Interpretation | Cluster 1 | Cluster 2 | Cluster 3 |
|---|---|---|---|
| Quiescence | 0.979 | 0.02 | $8.298 \times 10^{-5}$ |
| Dyke Propagation (S1-S4) | 0 | 0.029 | 0.971 |
| Eruption and presumed subglacial eruption | 0.178 | 0.810 | 0.012 |

Table 4: Confusion matrix between clusters formed by K-Means and the expert interpretation in Figure 4. The first row shows that 98% of the data points during quiescence are classified into cluster 1 and 2% are classified into cluster 2 and 3. The second row shows that 97% of the data points during the dyke propagation are classified into cluster 3, and 3% into cluster 2. The third row shows that 81% of the data points during the eruption and presumed subglacial eruption are classified into cluster 2, 12% are classified to cluster 2 and 1% are classified to cluster 3. The summation of values in each row is equal to 1.

[Figure]

Figure 5: Same as Figure 2 but with log(RMeS) instead of RMS. Cluster 1 is associated with quiescence, cluster 2 is associated with the eruption and presumed sub-glacial eruption, and cluster 3 is associated with dyke segments S1 to S4.

| K-Means
Expert Interpretation | Cluster 1 | Cluster 2 | Cluster 3 |
|---|---|---|---|
| Quiescence | 0.962 | 0.037 | $8.298 \times 10^{-5}$ |
| Dyke Propagation (S1-S4) | 0 | 0.036 | 0.964 |
| Eruption and presumed subglacial eruption | 0.122 | 0.865 | 0.013 |

Table 5: Confusion matrix between clusters formed by K-Means and the expert interpretation in Figure 5. The first row shows that 96% of the data points during quiescence are classified into cluster 1 and 4% are classified into cluster 2 and 3. The second row shows that 96% of the data points during the dyke propagation are classified into cluster 3 and 4% are classified into cluster 2. The third row shows that 87% of the data points during the eruption are classified into cluster 1, 12% are classified to cluster 2 and 1% are classified to cluster 3. The summation of values in each row is equal to 1.

---

## Author Response (AR2)

Dear Editor,

We would like to thank you for handling the review process of our manuscript. We also thank the reviewers for their valuable comments in improving the manuscript. Below is the summary of the manuscript changes with respect to the previous version:

1. We added new sub-sections in Method and Application to Seismic Data to introduce the concept and the settings of K-means clustering, which we used to assess the discriminative power of the quantified seismic parameters. This is according to the suggestion of Reviewer 1 to use an objective clustering algorithm.

2. We have revised section 5.5 Cluster analysis as suggested by both reviewers.

3. We have revised the Conclusion by taking account the suggestion from Reviewer 1.

4. We have revised the Abstract to better reflect the revised manuscript.

5. We have introduced the parameters RMS, RMeS and TADR in the Introduction as recommended by Reviewer 2.

6. We have improved the Figures using more distinctive colors that are still color-blind friendly.

7. We have also checked the references and corrected the false doi.

Please kindly find, in the next pages, our point-by-point replies to the reviewers' comments.

Kind regards,

The Authors

**POINT-BY-POINT REPLY**

*In the following, we comment point-by-point how the specific concerns and recommendations of the two referees have been met. Concerning each point, we refer to the place where the manuscript has been revised. The corresponding revisions are colored in the annotated manuscript.*

*Below, repetitions of the comments from the reviewers are written using black font, and the response for the authors are written using italic blue font.*

**Comments by Reviewer 1, anonymous,** https://doi.org/10.5194/egusphere-2024-1445-RC1

This manuscript contains a very careful analysis of dynamical changes of continuous seismic waveforms recorded during the diking episode and subsequent eruption of Bardarbunga volcano in 2014-2015. The analysis focuses on the use of Permutation Entropy and other useful metrics that can be applied even if only one station is available which is a common state of affairs in under-monitored volcanoes. The most interesting results of this work in my opinion are first, the good correlation of PE with discharge rate, and second, the great sensitivity that PE exhibits in detecting minute changes in the seismic wavefield due to temperature variations. I think that the manuscript is a good contribution to the field of volcano monitoring, it is clearly written, and it falls within the scope of the journal. I therefore recommend publication after the comments listed below are taken into account by the authors.

*__Response:__ We are very grateful to the reviewer for his positive statement and taking his time to provide useful feedback for the manuscript. Please find our responses to the specific comments below.*

1. The part of the manuscript on clustering of the estimated parameters is the weakest since it is subjective and it does not take into account clustering of other combinations of these parameters. Machine learning algorithms are widely used and software packages that implement them are readily available, hence I don't understand why the authors have not attempted to perform an objective analysis of clustering. If such algorithms are applied, then it will be possible to examine clustering beyond the binary (i.e two parameters at a time) way presented in the manuscript, but rather consider clustering in a higher-dimensional space (e.g., PE, PPE, IF, discharge rate etc).

*__Response:__ We thank the reviewer for this suggestion. Initially, the aim of the 'clustering' section was simply to compare the expert interpretation of the ongoing volcanic activity with the spatial position (here 2-D projections) in the parameter space. We achieved this by coloring the different eruption stages given by the expert classification. We hoped to see that data points of PE, PPE, and IF fall into different groups associated with the different eruption stages. However, it is a good suggestion to use an objective clustering algorithm. Now we added a clustering analysis using K-Means using different combination of parameters and presented the best one in __Figure 6__. We also calculated the confusion matrix, which quantifies how many points in each cluster formed by K-Means lies in the equivalent clusters formed by the expert interpretation, they are presented in Table S1 to S5 in the supplementary. We showed that using PE, PPE, IF and log(RMeS) gives a higher score in the confusion matrix. Therefore better in separating the eruption from the quiescence and the dyke propagation.*

2. At the end of the conclusions section the authors state that "...our study shows that it is still challenging to determine the eruption onset after the start of the pre-eruptive seismcity..."; I think that this statement may be accurate for the Holuhraun diking-eruption case but not for other studied

cases that utilized PE such as Gjalp or Shinmoedake cited by the authors. The main difference between Holuhraun and these past cases is the intense volcanotectonic seismicity prior to eruption due to the lateral propagation of the dike. The last part of the conclusions should be rephrased so that the reader becomes aware of this crucial difference and this methodology may be able to pinpoint the eruption onset when seismicity levels are lower.

**_Response:_** _This is a good point and we have revised the conclusion section accordingly. In the new conclusion, we also noted that the parameters may respond differently to each stage of the eruption. While one parameter may be more sensitive to one stage, the other responds better to another stage, so combining them may provide more reliable information._

3. In the supporting information Figures S3, and S5-S12 are plotted in a way that is difficult for the reader to easily follow the description of the authors in the Discussion section. I would suggest that these Figures are redrafted using different color for different lines by using a double y-axis, in the same way the authors plotted Figure 5 in the main manuscript.

**_Response:_** _Thank you for the feedback. We have revised the figures accordingly._

**Comments by Reviewer 2, anonymous,** https://doi.org/10.5194/egusphere-2024-1445-RC2

*Below, repetitions of the comments from the reviewer are written using regular font, and the response for the authors are written using italic font.*

The manuscript "Dynamical changes of seismic properties prior to, during, and after 2014-2015 Holuhraun Eruption, Iceland" presents some interesting results regarding the application of Permutation and Phase Permutation Entropy to continuous seismic waveform timeseries prior, during and after the eruption process. Changes in these measures seem to be associated with the dynamic changes due to magma propagation and eruption, which are also pointed by changes in the instantaneous frequency. Remarkably, Permutation and Phase Permutation Entropy also correlate well with the lava discharge rate. Overall, the manuscript presents some interesting results regarding volcanic monitoring, is well structured and written and falls within the scope of NHESS. Therefore, I recommend its publication after revising some points.
**_Response:_** *We are very grateful to the second reviewer for providing helpful suggestions for the manuscript. Please find our responses to the specific comments below.*

1) I believe that section 5.5 "Cluster analysis" is the most weak in the manuscript. The authors claim that by separating the various measures into clusters according to the pre-eruptive and eruptive processes, then these measures show good separation during the various stages. This seems true during the dike propagation stages that clearly separate from the quiescence and eruptive periods. However, the last two show significant overlapping particularly when Permutation and Phase Permutation Entropy are plotted (Fig.6a). Please elaborate more on this issue.

**_Response:_** *We thank the reviewer for the critical feedback. The clustering figure presented in the original manuscript was done according to an expert interpretation and was plotted on 2D-planes; therefore, two clusters may appear to overlap. In response to the similar suggestion from Reviewer 1, we have now added the K-Means clustering in the four-dimensional space and compare it to the expert interpretation. We have also added the Confusion Matrix to calculates how many points in each cluster formed by K-Means lies in the equivalent clusters formed by the expert interpretation. Please see the results in the attachment. We showed that using PE, PPE, IF and log(RMeS) gives a higher score in the confusion matrix. Therefore better in separating the eruption from the quiescence and the dyke propagation. This part will be added to revised section 5.5.*

2) Remove duplicate words from the text, as for instance in lines 27 and 408.

**_Response:_** *We have rephrased those sentences to avoid duplicate words.*

3) Although these are explained further in the text, introduce in line 78 what RMS, RMeS and TADR stand for, as this is the first time that are encountered.

**_Response:_** *Thank you, we now introduced these parameters in the Introduction section*

4) The colors for S1 and S2 in Fig.6 are not clearly distinctive.

**_Response:_** *We have revised the figure using more distinctive colors that are still color-blind friendly.*

5) Check your references, as doi for some seem to refer to other publications in the list.

**_Response:_** *Thank you for your careful and detail attention. We have checked and corrected the incorrect DOIs.*